# Regulation Mechanisms of Meiotic Recombination Revealed from the Analysis of a Fission Yeast Recombination Hotspot *ade6-M26*

**DOI:** 10.3390/biom12121761

**Published:** 2022-11-26

**Authors:** Kouji Hirota

**Affiliations:** Department of Chemistry, Graduate School of Science, Tokyo Metropolitan University, Minamiosawa 1-1, Hachioji-shi, Tokyo 192-0397, Japan; khirota@tmu.ac.jp

**Keywords:** meiotic recombination, double strand break (DSB), Rec12 (fission yeast Spo11 homolog), chromatin, transcription factor, *S. pombe*

## Abstract

Meiotic recombination is a pivotal event that ensures faithful chromosome segregation and creates genetic diversity in gametes. Meiotic recombination is initiated by programmed double-strand breaks (DSBs), which are catalyzed by the conserved Spo11 protein. Spo11 is an enzyme with structural similarity to topoisomerase II and induces DSBs through the nucleophilic attack of the phosphodiester bond by the hydroxy group of its tyrosine (Tyr) catalytic residue. DSBs caused by Spo11 are repaired by homologous recombination using homologous chromosomes as donors, resulting in crossovers/chiasmata, which ensure physical contact between homologous chromosomes. Thus, the site of meiotic recombination is determined by the site of the induced DSB on the chromosome. Meiotic recombination is not uniformly induced, and sites showing high recombination rates are referred to as recombination hotspots. In fission yeast, *ade6-M26*, a nonsense point mutation of *ade6* is a well-characterized meiotic recombination hotspot caused by the heptanucleotide sequence 5′-ATGACGT-3′ at the *M26* mutation point. In this review, we summarize the meiotic recombination mechanisms revealed by the analysis of the fission *ade6-M26* gene as a model system.

## 1. Introduction

Meiosis is a special cell division process that creates gametes by reducing the number of chromosomes by half to maintain the original chromosome number during sexual reproduction. Meiosis consists of DNA replication followed by two sequential rounds of nuclear division: MI and II. Homologous chromosomes segregate away from each other in the first nuclear division (MI), whereas the sister chromatids separate in the second meiotic division (MII) [1]. Before MI, the two homologous chromosomes are in close contact to form bivalent chromosomes, which is pivotal for the faithful segregation of homologous chromosomes [2]. This contact is mediated by crossover (CO) (cytologically seen as chiasma) formed between two homologous chromosomes due to homologous recombination [3,4,5]. 

Meiotic homologous recombination contributes to the proper segregation of homologous chromosomes and augmentation of genetic diversity. This is a highly regulated process initiated by DNA double-strand breaks (DSBs), which are induced by the conserved Spo11 protein and other accessory DSB proteins including Rec114 and Mer2, which assemble on chromosomes [6,7,8,9,10,11]. Spo11 is an enzyme similar to topoisomerase II and induces DSB via nucleophilic attack of the hydroxy group in its catalytic residue tyrosine (Tyr) on the phosphodiester bond [5,12] (Figure 1A,B, DSB formation). The programmed DSBs are repaired by homologous recombination using the homologous chromosome donor sequence (Figure 1B). DSB ends are processed into 3′ single-stranded overhangs via end resection by the CtIP, Exo1, and Mre11-Rad50-Nbs1 (MRN) complex [13,14,15,16] (Figure 1B, End resection). The resultant single stranded DNA (ssDNA) can invade homologous chromosomes via homologous recombination [17,18] (Figure 1B). From bacteria to humans, most types of homologous recombination rely on the RecA family of recombination proteins (eukaryotic homologs, Rad51, and Dmc1) to promote strand pairing and invasion [17,19]. This view is supported by the facts that mutations in *DMC1* gene cause infertility in human and mouse [20,21]. Each of these recombination proteins forms a nucleoprotein filament by binding to 3 nucleotides of the ssDNA [22]. The resultant nucleoprotein filament helps identify a homologous genome sequence within the donor double-stranded DNA (dsDNA) sequence and then promotes an exchange of base pairs, leading to the formation of a displacement loop (D-loop) [23]. After invasion and exchange of the base pairs, a DNA polymerase extends the 3′ end of the invading strand using a homologous template and eventually repairs the DSB. If the synthesized DNA end returns and rehybridizes to the original strand, a portion of the sequence in the homologous chromosome is copied. This process is referred to as synthesis-dependent strand annealing (SDSA) and results in gene conversion (non-crossover [NCO]) (Figure 1B, SDSA). Alternatively, annealing between the displaced strand and complementary ssDNA (second-end capture) results in the formation of a double Holliday junction (dHJ), leading to CO [24,25] (Figure 1B, dHJ). This dHJ model was proposed from studies in *Saccharomyces cerevisiae* (budding yeast), while other model in which COs are induced from single Holliday junction (sHJ) was proposed in the highly divergent yeast *Schizosaccharomyces pombe* (fission yeast) [26]. In this model, the second-end capture results in the formation of sHJ, which is resolved by Mus81 resulting in CO or NCO [26] (Figure 1B, sHJ). 

The COs/chiasmata and meiotic DSBs are non-randomly distributed along the chromosome and concentrated in short DNA intervals called DSB hotspots [27,28,29,30]. The global distribution of DSB has been extensively studied in budding yeast and fission yeast [31,32,33,34]. Nucleotide resolution analysis of DSB sites using next-generation sequencing in budding yeast has provided insights into DSB sites and hierarchical contexts, such as chromosome structures, chromatin, transcription factors, and local sequence composition [35]. Many of the DSB hotspots are located in promoter regions for mRNA or noncoding RNA (ncRNA) [36,37], which are characterized by transcription factor binding, histone modifications, and low nucleosome density, suggesting a close relationship between meiotic recombination and transcription. Chromatin consists of arrays of nucleosomes, which are one of the critical determinants of all bioreactions on chromosomes, such as transcription, replication, and recombination, with the following correlations: open chromatin configuration correlates with active reactions and closed (condensed) chromatin configuration correlates with inactivation [38,39]. Moreover, the involvement of the state of chromatin figurations and histone variant in the control of CO/NCO has been shown in the genome wide researches in arabidopsis (reviewed in [40]). In addition to chromatin status, the status of mRNA and ncRNA transcription may also be critical determinants of meiotic recombination [36,37]. For several decades, researchers have explored the multiple layers of meiotic recombination modulators by analyzing the fission yeast gene *ade6-M26* as a model meiotic recombination hotspot. Herein, we discuss how analyses of the fission yeast gene *ade6-M26* have provided knowledge on the multiple regulatory mechanisms involved in the regulation of meiotic recombination.

## 2. *ade6-M26*, a Meiotic Hotspot Created by a Mono-Nucleotide Substitution Mutation

The *ade6* gene encodes phosphoribosylaminoimidazole carboxylase, which is required for purine biosynthesis. The adenine auxotrophic strains with a nonsense mutation in the *ade6* gene form red (or pink) colonies on yeast extract medium with a limiting adenine concentration [41], providing a visual distinction between wild-type and mutant alleles (Figure 2A). Several hundred *ade6* mutations induced by ultraviolet or X-ray irradiation or nitrous acid treatment were assessed for recombination to map the mutation point, and the mutation *ade6-M26* was shown to create a meiotic recombination hotspot [42]. The *ade6-M26* hotspot contains a G-to-T substitution in the *ade6* open reading frame (ORF) [43,44] (Figure 2B, Red letter). During the early stage of genetic analysis in meiotic recombination, the *rec* genes required for meiotic recombination were screened using this system and several *rec* genes were identified [45,46]. Originally, *ade6-M26* hotspot was identified as a gene conversion (NCO) hotspot [42] (Figure 1B, left SASD). Later, it was showed that this mutation hotspot also augments the meiotic CO recombination [47] (Figure 1B, left Crossover).

## 3. Transcription Factor Required for Meiotic Recombination at *M26*

The *M26*-mutation is a nonsense mutation that creates a cyclic adenosine monophosphate (cAMP)-responsive element (CRE)-like heptanucleotide sequence 5′-ATGACGT-3′ (Figure 2B, Red square). Later, biding factors to this CRE-like heptanucleotide sequence was identified and named as Mst1 and Mst2 and it was revealed that these bind factors activate meiotic recombination [48,49,50] (Figure 2C). After this identification, three groups almost simultaneously identified Atf1, a basic leucine zipper (bZIP) transcription factor that is activated by the mitogen-activated protein kinase (MAPK) pathway [51,52,53]. Moreover, a similar bZIP protein, Pcr1, was also identified [54], and Mts1 and Mts2 were found to be identical to Atf1 and Pcr1, respectively [48]. In fission yeast, meiosis is induced by the depletion of the nitrogen source, and the signal is mediated by PKA and stress-activated MAPK signal pathway, resulting in the activation of the genes required for meiosis [55]. In addition, during nitrogen starvation, the phosphorylation of Atf1 through a stress-activated MAPK, Spc1, is required for conjugation and meiosis [52]. Atf1 phosphorylation is also required for the activation of the *fbp1* gene upon glucose starvation stress via the modulation of chromatin configuration in *fbp1* upstream sequence [56,57,58,59,60,61,62,63]. Thus, Atf1 phosphorylation is involved in *ade6-M26* hotspot activation, gene induction during mating and meiosis, and *fbp1* transcription. Interestingly, the canonical Spc1-phospholylation sites in Atf1 are required for *fbp1* transcription but not for the activation of the *ade6-M26* hotspot or induction of meiotic genes, indicating a distinctly different phosphorylation mechanism required for *ade6-M26* hotspot activation, gene transcriptional induction in mating and meiosis, and *fbp1* transcription [64]. Similarly, a meiotic recombination hotspot, *ade6-3049* allele possesses a point mutation, which concurrently creates a nonsense mutation and a CRE-like sequence at the 3′ region of *ade6* gene [37]. Both *ade6-M26* and *ade6-3049* strains express shorter versions of mRNA initiated from *M26* and *3049* point mutations, respectively [37,65] (Figure 2C), suggesting a link between transcriptional initiation and meiotic recombination. Involvement of transcriptions in the meiotic recombination activation at *ade6-M26* is also highlighted by the following facts. First, hotspot activity of *ade6-M26* requires *ade6* promoter [66,67]. Second, insertions of other transcription factor binding sequences at the *M26* mutation site in *ade6* gene also created meiotic recombination hotspot [68,69]. This indicates that meiotic recombination hotspots can be created by transcription factors other than the Atf1-Pcr1 transcription factor required for the *ade6-M26* hotspot [48,49,50]. This was proved to be the case when the loss of each transcription factor bound to the inserted binding site inactivated hotspot activity [70]. Moreover, the subset of naturally occurring *M26* sequence sites (*M26* heptanucleotide sequence occurring in wild-type cells) that facilitate meiotic recombination is implicated in approximately 20% of all natural meiotic recombination events in the fission yeast genome [71]. This might also be the case in another highly diverged yeast, *Saccharomyces cerevisiae* (budding yeast), since binding sites for the Atf1 ortholog Sko1 tend to be highly enriched for meiotic DSBs [35]. Furthermore, other sequence-specific binding proteins proven to activate hotspots in the *ade6* gene in fission yeast (Php2, Php3, Php5, Rst2) [68,69] also have orthologs in budding yeast (Hap2, Hap3, Hap5, Adr1) in which DSBs are preferentially directed to their binding sites [35]. Such sequence-specific transcription factor-dependent hotspots created in the fission yeast *ade6* gene are mediated through a common chromatin modulation system [72] (see Section 4). Taken together, these results suggest that some shared regulatory systems between recombination and transcription may play a role in a subset of meiotic recombination sites created by the binding sequence of transcription factors. 

## 4. Chromatin Modulation Mechanisms Involved in the Regulation of *ade6-M26* Meiotic Recombination Hotspot 

The chromatin configurations play pivotal role in the transcriptional control (reviewed in [38,39]). This can be attributed to the fact that tightly packed chromatin limits the accessibility of trans-acting proteins such as transcription factors. Regulation of chromatin structure involves two major classes of chromatin modulation machinery. The first comprises histone modifications, such as acetylation, that are involved in chromatin modulation. For example, histone acetyltransferases add acetyl groups that are usually associated with active chromatin [73,74]. The second class comprises ATP-dependent chromatin remodelers. ATP-dependent chromatin remodelers alter chromatin configuration by sliding and disassembling nucleosomes in an ATP-dependent manner [75]. The yeast Swi2/Snf2 family ATP-dependent chromatin remodeler contains a helicase/ATPase motif as a catalytic domain and a bromodomain that recognizes acetylated histones [76,77]. Proteins carrying bromodomains, including Swi2/Snf2 family ATP-dependent chromatin remodelers, can serve as decoders for histone acetylation. 

The direct involvement of histone acetylation and ATP-dependent chromatin remodeler-mediated chromatin remodeling in the regulation of meiotic recombination was first demonstrated in *ade6-M26* [78]. The chromatin configurations around the *M26* mutation site were assessed based on the sensitivity of the chromatin DNA to micrococcal nuclease (MNase). The chromatin configuration at the *M26* mutation point was shown to convert into an open state during meiosis, which requires the involvement of Atf1, Pcr1, and CRE-like heptanucleotide sequences [79] (Figure 3A). Acetylation of the histones H3 and H4 around the *M26* mutation site is augmented in an *M26*- and Atf1/Pcr1-dependent manner early in meiosis. Gcn5 histone acetyl transferase is involved in the histone H3 acetylation around *M26* [78]. Moreover, deletion of the *gcn5* gene results in a significant delay in chromatin remodeling and partial reduction in the *M26* meiotic recombination frequency [78]. Deletion of the *snf22* gene (which encodes Swi2/Snf2-chromatin remodeler) abolishes chromatin remodeling and results in the critical reduction of meiotic recombination at *ade6-M26* site [78]. These results suggest that Gcn5 histone acetyl transferase and Snf22 Swi2/Snf2-chromatin remodeler cooperatively alter the local chromatin structure to activate meiotic recombination at *M26* in a site-specific manner (Figure 3B). Later, two CHD-1 family chromatin remodelers (Hrp1 and Hrp3) involved in chromatin regulation, a Gcn5 histone acetyltransferase component (Ada2), a member of the Moz-Ybf2/Sas3-Sas2-Tip60 family histone acetyltransferase (Mst2), and a histone variant H2A.Z have been reported in the regulation of the of *ade6-M26* hotspot activity [80,81]. Moreover, proteomic analysis of the *ade6-M26* recombination hotspot revealed that chromatin-mediated regulators, including histone chaperones (Nap1, Hip1/Hir1), subunits of the Ino80 complex (Arp5, Arp8), a DNA helicase/E3 ubiquitin ligase (Rrp2), components of a Swi2/Snf2 family remodeling complex (Swr1, Swc2), and a nucleosome evictor (Fft3/Fun30), interact with the *ade6-M26* recombination hotspot during meiosis [82]. Taken together, these studies revealed that remarkably diverse chromatin modulators and histone post-translational modifications participate in the regulation of *M26* meiotic recombination.

## 5. Meiotic Double-Strand Break Formation around *M26* Mutation Site

Meiotic recombination is thought to be initiated by either single-strand breaks or double-strand breaks in the DNA [83,84]; however, which of the two is specifically responsible has been a long-standing question. This controversy was resolved by physical detection of double-strand breaks (DSBs) in budding yeast during meiosis [85]. The formation of DSBs in budding yeast requires multiple gene products that are also required for meiotic recombination and proper homologous chromosome segregation [86]. One such gene product, Spo11 is required for the formation of meiotic DSBs [87]. The fission yeast Spo11 homolog, Rec12 protein, is also essential for meiotic recombination [45], and thus this protein is hereafter referred to as Spo11/Rec12. The catalytic tyrosine residue of Spo11/Rec12 is pivotal for its action, suggesting that the DNA–Spo11/Rec12 covalent bond is mediated by this tyrosine residue [88,89] (Figure 1A). These results suggest a conserved mechanism for meiotic DSB formation and recombination. However, the initial attempt to assess meiotic DSBs in the fission yeast genome suffered due to a lack of a detectable level of DSBs at or near the *ade6-M26* hotspot [89]. The unresolved question whether DSBs are associated with the *ade6-M26* hotspot was resolved by using the *rad50s* mutant. This mutant, originally obtained from budding yeast, was found to be defective in end resection [90,91], and covalent Spo11/Rec12-DNA complexes were detected in this mutant [87,92] (DSB formation and end resection step in Figure 1B). In fission yeast, the *rad50s* mutant also shows accumulation of DSBs and no detectable repair [93]. Multiple DSBs were detected on both sides near the *M26* mutation site (5′-ATGACGT-3′ heptanucleotide sequence) with closely spaced *M26* sites [94]. Interestingly, the *M26* site itself lies in a region of approximately 70 bp that is free of visible breaks (Figure 4), suggesting that Spo11/Rec12 is recruited to the genomic DNA where the chromatin configuration has been meiotically opened by the Atf1-Pcr1 and stochastic digest the surrounding sequences.

## 6. Induction of Meiotic DSB via the Expression of Long ncRNA

DSBs induced by Spo11/Rec12 are not uniformly distributed in the genome [87,92,93], and a significant overlap between the DSB sites and the regions of ncRNA expression was identified by bioinformatics analysis in fission yeast [36]. These facts suggest that the accessibility of Spo11/Rec12 to the genome could be controlled by ncRNAs. In this section, the role of ncRNA transcriptions in the induction of meiotic DSB is discussed.

The ncRNAs can be classified into two classes, small RNAs and long ncRNAs (lncRNAs), according to their length. The first class of ncRNAs, small RNAs are ncRNAs comprised of less than 200 nucleotides. siRNAs and miRNAs is well characterized examples of small RNAs. Small RNAs regulate gene expression via degradation of mRNA and inhibition of translation through their interaction with RNA-induced silencing complexes [95]. The second class of ncRNAs is long ncRNA (lncRNA) carrying over 200 nucleotides length. They are involved in various biological processes, including gene regulation, development, nuclear organization, and cancer development [96,97]. Deregulation of several lncRNAs in cancer cells has been identified and such lncRNAs can be used as a cancer marker in the diagnosis [98,99]. The expressed lncRNAs may serve as enhancers, scaffolds, or decoys by interacting with other RNAs or proteins, resulting in the aberrant activation of genes required for cancer proliferations [100]. In addition to the roles played by transcribed lncRNA molecules, several studies have suggested the importance of *cis*-acting RNA polymerase II-transcribing nascent lncRNAs in the regulation of neighboring genes through their effects on histone modifications [101,102]. The lnc RNAs expressed from gene promoter region are referred to as promoter-associated lncRNA, which are categorized into most abundant ncRNA, and play pivotal roles in the regulation of downstream genes [103]. 

The fission yeast *fbp1* gene has been extensively analyzed for understanding gene regulation systems (reviewed in [104]), and several promoter-associated lncRNAs are identified in its promoter region [59,105]. These lncRNAs are referred to as metabolic stress-induced long-ncRNA (mlonRNA) and are expressed from specific *cis*-element, mlon-BOX, before induction of *fbp1* mRNA upon glucose starvation stress [106,107]. Transcriptional initiation of mlonRNAs modulates chromatin structure around the *fbp1*-promoter and converts it into an open configuration, thereby promoting the binding of transcription factors to the promoter [59]. A significant overlap between the *mlon-BOX* and meiotic DSB sites in the fission yeast genome suggests interesting hypothesis that mlonRNA transcriptional initiations determine the DSB site via the modulation of chromatin architecture [65,108]. This hypothesis was assessed by inserting the *mlon-BOX* sequence into the *ade6-M26* hotspot. To mimic *fbp1* regulation, in which the *mlon-BOX* sequence is located 200 bp downstream from the Atf1 binding site in *fbp1*, the *mlon-BOX* sequence was placed 200 bp downstream from the *M26* mutation site in *ade6-M26* (Figure 5A,B). This insertion caused additional activation of transcription at the insertion site (Figure 5B) and stimulated meiotic recombination via local chromatin remodeling [65] (Figure 5C). These results suggest that mlonRNA transcription plays a universal role in chromatin remodeling and the regulation of transcription and recombination. Taken together with the regulatory mechanisms discussed in earlier sections, following model can be illustrated. The accessibility of Spo11/Rec12 to the genome could be one of the critical determinants of DSB sites and that Spo11/Rec12 might preferentially make DSBs at genomic regions showing open chromatin configurations induced by *cis*-acting RNA polymerase II transcribing nascent mlonRNAs. 

## 7. Summary and Perspective

In this review, we summarized the multiple layers of regulation involved in meiotic recombination revealed from the analysis of the fission yeast *ade6-M26* hotspot as a model system. Analyzed regulatory layers include transcription factors, chromatin modulators (histone modifiers and ADCRs), and lncRNA transcriptions. Although intensive analyses have been conducted using this model system, several important questions remained unsolved. The first question is what the proportion of meiotic recombination sites is dependent on transcription factors. The subset of naturally occurring *M26* (binding sequence for the Atf1 transcription factor) sites that facilitate meiotic recombination supposedly mediate approximately 20% of all meiotic recombination events in the fission yeast genome [71]. The chromatin structure at prominent natural hotspots, such as major break site 1 (*mbs1*) in fission yeast, shows a constitutive open configuration independent of Atf1 [109,110]. These constitutive hotspots might be important for ensuring the basal recombination level required for CO, which is essential for the progression of meiosis. In contrast, recombination sites linked to transcription factor-binding sequences, such as *M26* sites, might be important for the plasticity of meiotic recombination hotspots. Environmental conditions and stresses also affect the frequency and distribution of meiotic recombination [111]. The binding patterns of the transcription factors associated with meiotic recombination could possibly be affected by environmental conditions and thereby affecting the frequency and distribution of recombination. This was proved to be the case when hotspots created in the *ade6* gene carrying a binding sequence for different transcription factors such as Atf1-Pcr1, Rst2, or Php2-Php3-Php5 were affected differently by distinct types of stresses [70]. The second unsolved question involves the factor(s) and/or conditions required for creating *M26* sequence hotspots. The *ade6* gene has a CRE sequence (5′-TGACGTCA-3′) at +1131 to +1136 bp starting from the first A in the *ade6* ORF. However, Atf1-Pcr1 could not bind to this CRE sequence and could not facilitate recombination. Consistently, little meiotic DSB is introduced at CRE sequence in the regulatory region of *fbp1* gene (our unpublished observation), indicating that not the all CRE sequences in fission yeast genome are meiotic recombination hotspot. Similarly, the *HIS4* locus in budding yeast was initially shown to be linked to the activity of the transcription factors Bas1 and Bas2 [112], but subsequent genome-wide analyses failed to detect a clear correlation between Bas1/2 activity and DSB formation [113]. Thus, other factor(s) and/or condition(s) may be involved in making these transcription factors-binding sites meiotic recombination hotspot. Further research is required to clarify these questions. 

## Figures and Tables

**Figure 1 biomolecules-12-01761-f001:**
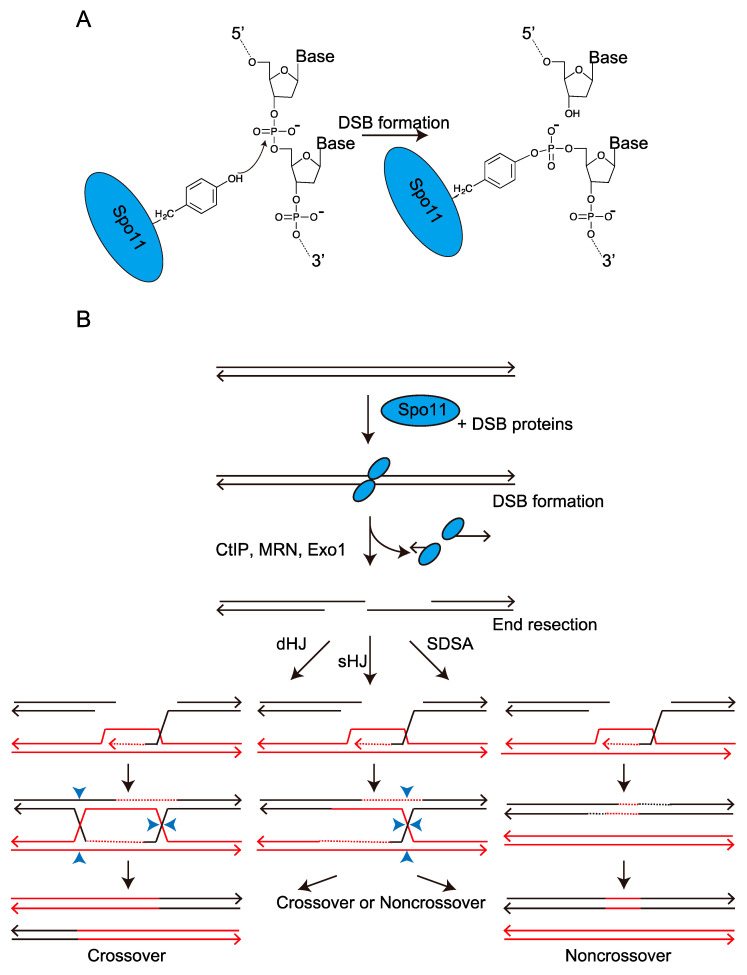
Meiotic double-strand break (DSB) formation and homologous recombination induced by the programed DSBs. (**A**) Mechanism of the DSB formation catalyzed by Spo11 protein. The tyrosine (Tyr) side chain on the Spo11 protein carries out nucleophilic attack on the DNA phosphodiester backbone. This transesterase reaction cuts the DNA backbone and covalently links the Spo11 protein to the DNA end via a tyrosyl phosphodiester linkage. Spo11 protein is represented by a blue circle. (**B**) Schematic representation of meiotic recombination. Spo11 protein is represented by a blue circle. Homologous donor DNA is indicated by the red line. Only one of the two sister chromatids from each homolog has been shown.

**Figure 2 biomolecules-12-01761-f002:**
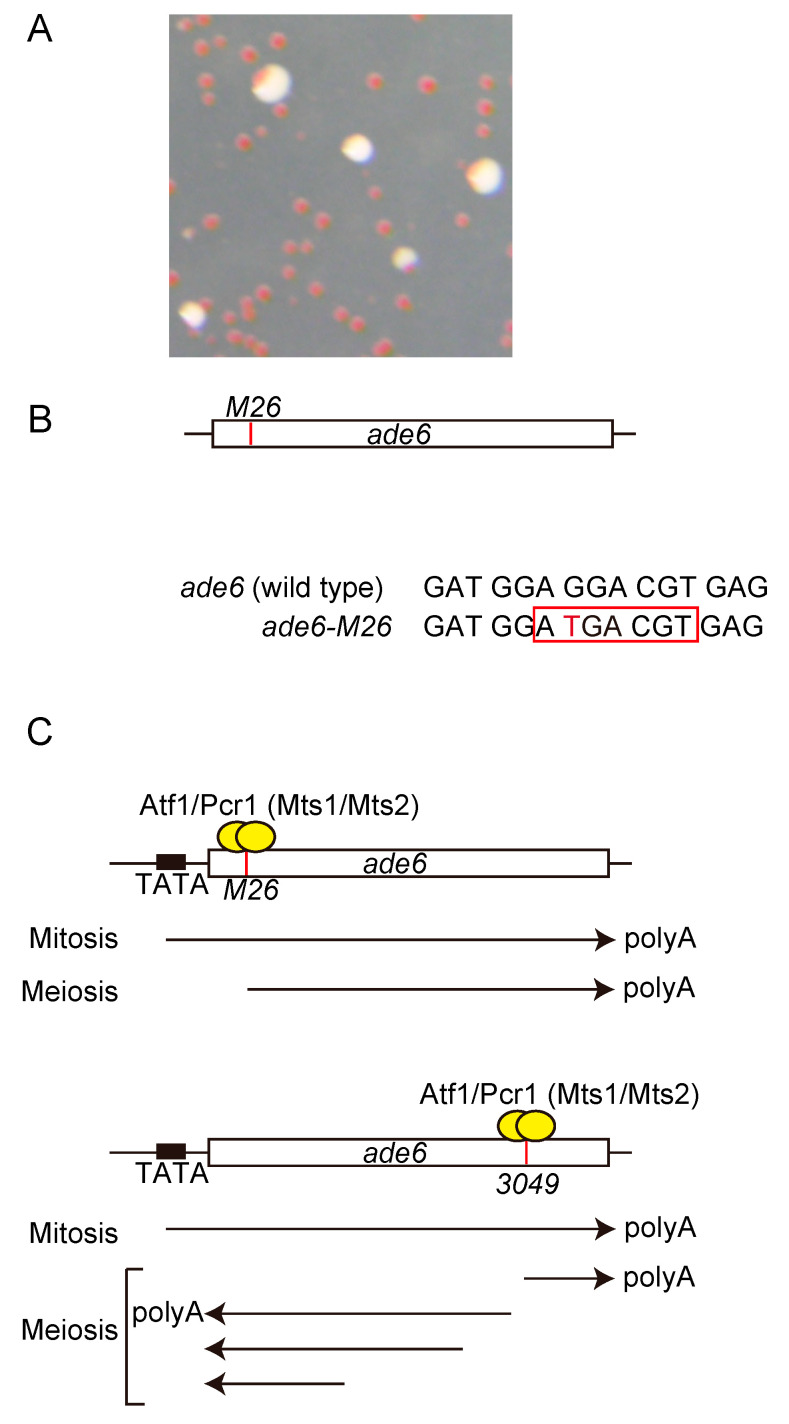
*ade6-M26* meiotic recombination hotspot created by single nucleotide substitution. (**A**) Image of pink colonies on a medium containing limited adenine concentration. (**B**) Schematic representation of *ade6-M26* allele. The white box shows *ade6* gene. The red vertical line shows the *M26* mutation point. The sequences around *M26* mutation are indicated. The red letter indicates the single nucleotide substitution in the *ade6-M26* allele. The red square shows cyclic adenosine monophosphate (cAMP)-responsive element (CRE)-like heptanucleotide sequence. (**C**) Schematic representation of *ade6-M26* and *ade6-3049* alleles. White boxes show *ade6* gene. Red vertical lines show each mutation. Yellow circles show Atf1 and Pcr1 (Mst1 and Mts2) transcription factors. Arrows indicate transcripts. In mitosis, *ade6-M26* and *ade6-3049* strains express *ade6* gene from TATA box, while the transcriptional initiation site was shifted to Atf1/Pcr1 binding sites in meiosis.

**Figure 3 biomolecules-12-01761-f003:**
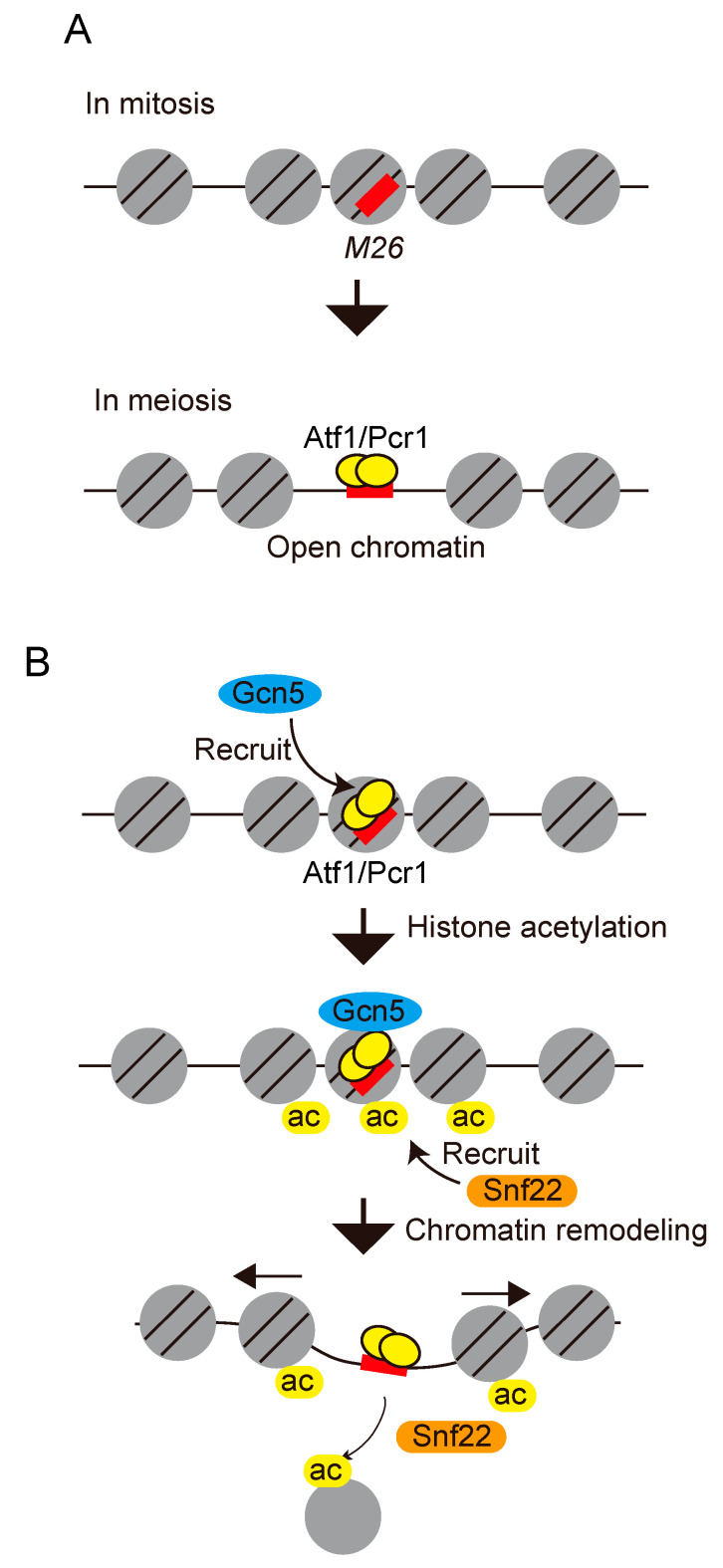
A model of meiotic recombination at *ade6-M26* by histone acetylation-mediated chromatin remodeling. (**A**) During meiosis, the state of the chromatin at the *M26* mutation site changes to open configuration. This chromatin alteration is dependent on Atf1/Pcr1 and CRE-like heptanucleotide sequence, 5′-ATGACGT-3′. The binding of Atf1 to *M26* site is enhanced in meiosis. (**B**) Histones around *M26* are acetylated by Gcn5 histone acetyl transferase and probably other histone acetyl transferases, which are recruited to the *M26* site via Atf1/Pcr1. Acetylated histones are recognized by ATP-dependent chromatin remodelers including Snf22. DSB machinery is recruited to the open chromatin region created around *M26*, thereby activating meiotic recombination.

**Figure 4 biomolecules-12-01761-f004:**
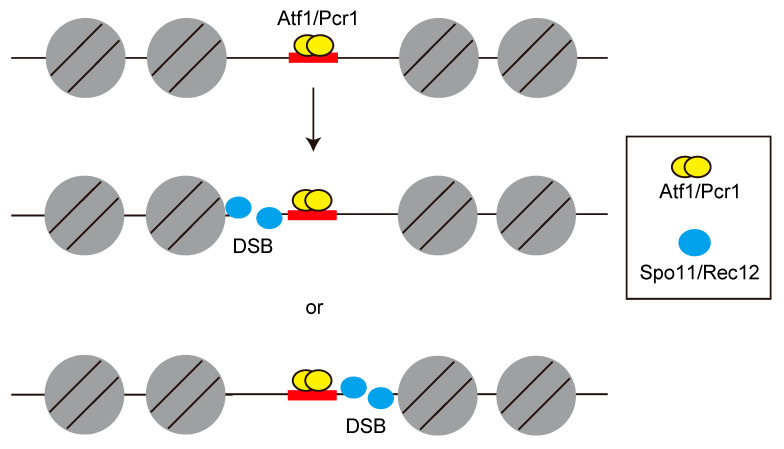
Schematic representation showing the position of DSBs within *ade6* gene. The red box indicates the position of the CRE-like heptanucleotide sequence generated due to *M26* mutation. DSBs were detected on both sides near the *M26* mutation site surrounded by closely spaced *M26* sites, whereas the *M26* site itself lies in a region free of visible breaks.

**Figure 5 biomolecules-12-01761-f005:**
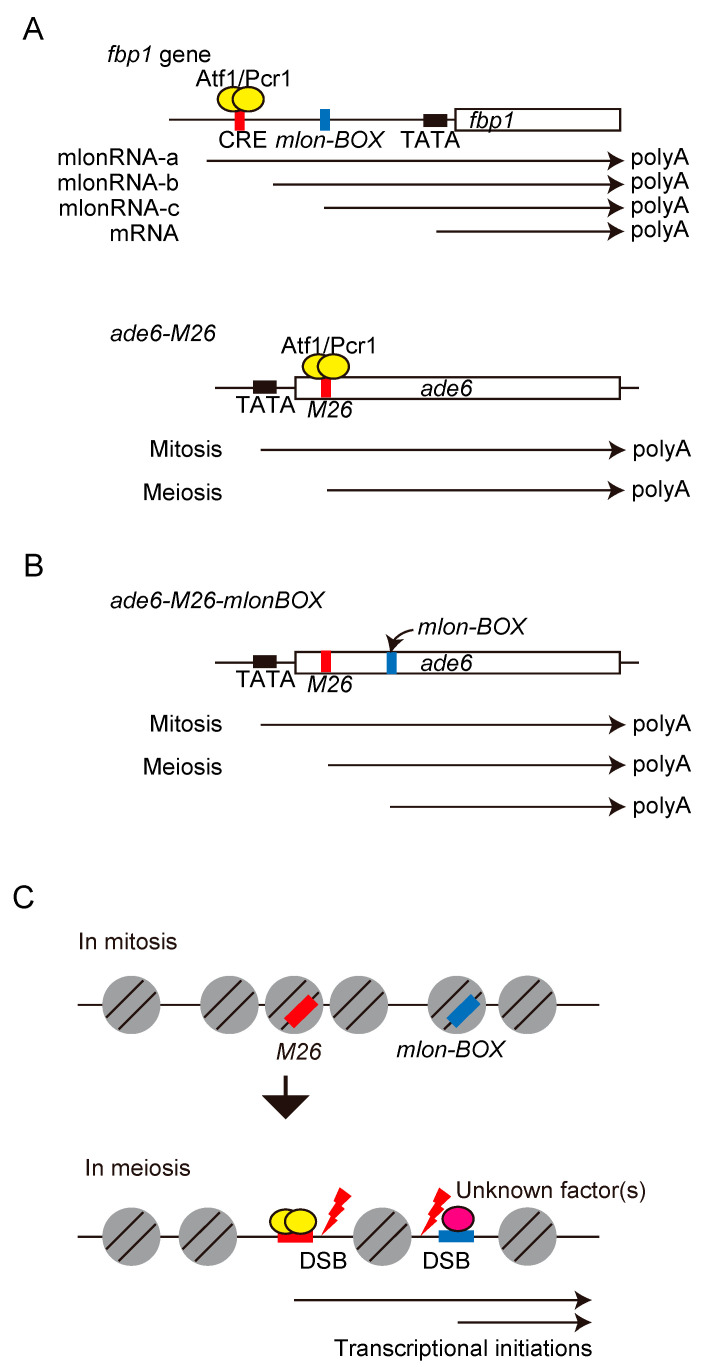
The conserved role of promoter-lncRNAs, mlonRNAs identified in the upstream region of *fbp1* during the regulation of chromatin. (**A**) Schematic representation of mlonRNA transcriptions from the *fbp1* and *ade6* loci. In the *fbp1* locus, during glucose rich condition, the longest mlonRNA (mlonRNA-a) is weakly expressed. Initially, during glucose starvation (10–20 min of glucose starvation), mlonRNA-b and -c are progressively expressed. At 60–180 min of glucose starvation, *fbp1*-mRNA is massively induced. *mlon-BOX* (mlonRNA-c initiation element), is located approximately 100 bp upstream from mlonRNA-c transcription start site. Notably, *mlon-BOX* is around 200 bp downstream from the Atf1 binding site (CRE). Red, blue, and black boxes show CRE, *mlon-BOX*, and TATA-BOX, respectively. In the *ade6* locus, during mitotic cell phase, mRNA is expressed from TATA-BOX. During meiosis, the transcription initiation site is shifted to the CRE site. (**B**) Schematic representation of *ade6* gene locus. Position of TATA-box, *M26*, and the insertion point of *mlon-BOX* are indicated by black, red and blue boxes, respectively. The *ade6* mRNA and transcripts induced from the *M26* site or *mlon-BOX* are indicated by arrows. (**C**) Chromatin configuration around *M26* site and *mlon-BOX* insertion point. Transcriptional activations occur form *M26* site and *mlon-BOX* insertion point in meiosis. Then, chromatin configuration changes to open state and DSBs are induced at the *M26* site and *mlon-BOX* insertion point. The DSB sites are shown.

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
