# Peer review of "Regulation Mechanisms of Meiotic Recombination Revealed from the Analysis of a Fission Yeast Recombination Hotspot ade6-M26"

_biomolecules, 2022, doi:10.3390/biom12121761_

Round 1

Reviewer 1 Report

At the early stages of meiosis, programmed formation of DNA double-strand breaks (DSBs) across the genome induces meiotic recombination. In this manuscript, Hirota reviews our current knowledge of meiotic DSB formation at a very well characterized meiotic DSB hotspot in fission yeast. In particular, the review focuses on the role of transcription and chromatin regulation in promoting DSB formation at this locus.

The review is overall well written and timely, but I think some edits could make this interesting topic more generally accessible. In addition, some sections feel overly repetitive and could benefit from edits that remove redundancies.

1.     The link between transcription and DSB formation is indeed very strong at the ade6-M26, but I think the reader would benefit from highlighting that transcription does not always have a very clear causal relationship with DSB formation. For example, the HIS4 locus in budding yeast was initially shown to be very clearly linked to the activity of the transcription factors Bas1/2 but subsequent genome-wide analyses failed to observe a clear correlation between Bas1/2 activity and DSB formation.

2.     The discussion of ncRNA (section 6) should be tied better into the review. The relevance of these RNAs is not particularly well defined and the discussion of the data is quite technical. I suggest streamlining the discussion of the primary data and putting this data more into context with respect to the other aspects of ade6-M26 regulation discussed earlier. Also, how general is the role of ncRNAs in DSB formation? Some discussion of the roles of ncRNAs in DSB formation in other systems could be helpful to provide context.

3.     Several pieces of information are presented multiple times, creating unnecessary redundancies. For example, the catalytic mechanism of Spo11/Rec12 is described three times (including in the abstract). Similarly, reintroduction of transcriptional effects at the beginning of section 6 seems unnecessary.

4.     I suggest removing researchers’ names from the text to keep the focus on the science. The researchers’ contributions are readily seen in the citations.

5.     In general, abbreviations should only be introduced for terms that are used again later. I may have missed it, but abbreviations like NGS and HAT do not seem to be used after their introduction. I also wonder whether less common abbreviations such as ADCRs are  necessary. If there is no character limit then I think using “chromatin remodelers”, rather that ADCRs, throughout the later mentions would greatly help readability.

6.     ade6-3049: is this allele a DSB hotspot? Some of the later discussion seems to imply this but this is not spelled out when the allele is introduced.

7.     Clarify the term “naturally occurring M26 sequence sites”. At first read, I thought this meant that some natural S. pombe isolates encode the ade6-M26 allele. Later in the review it becomes clearer that this term refers to other instances of this sequence motif in the S. pombe genome.

8.     Figure 3B: Why are Atf1/Pcr1 omitted in the last panel? Is there any evidence the two proteins ever leave?

9.     Figure 4 and last lines in section 5: I am not convinced that there is any evidence for stable binding of Spo11/Rec12 to the DSB site. The observed lack of cleavage at the M26 sequence could simply be the result of a footprint by Atf1/Pcr1 that blocks access for the Spo11/Rec12 nuclease.

10.  The author makes a good attempt to simplify nomenclature by using the most commonly used gene names across organisms (e.g. Spo11 vs Rec12). However, for Spo11/Rec12 this nomenclature is not use consistently, and Red12 is introduced as the homolog of Spo11 twice. I think the species-specific nomenclature could be introduced earlier, using Rec12 from then on out. Alternatively, using a double name such as Spo11/Rec12 or Spo11 (Rec12) consistently could help avoid confusion.

Minor points:

1.     When talking about Spo11/Rec12, please clarify that the similarity is specifically with topoisomerase II (or topoisomerase II-like enzymes), rather than topoisomerase in general.

2.     Tyr should also be written out as tyrosine at first mention.

3.     Typo: Introduction, line 3: “Meiosis consists of DNA replication followed by two….”

4.     Introduction, paragraph 1, last line: crossovers and chiasmata should be plural in this sentence

5.     Introduction, paragraph 2, second and third sentences from the end: “resulting in gene conversion” and “results in gene conversion” is repetitive. I suggest deleting this clause in the first of the two sentences.

6.     Introduction, paragraph 2, last line: in S pombe there is clear evidence for single Holliday junctions. I suggest rephrasing this part and maybe referring to joint molecules (as a more general term) and then clarifying that these can be either double or single-Holliday junctions

Author Response

Reviewer 1

At the early stages of meiosis, programmed formation of DNA double-strand breaks (DSBs) across the genome induces meiotic recombination. In this manuscript, Hirota reviews our current knowledge of meiotic DSB formation at a very well characterized meiotic DSB hotspot in fission yeast. In particular, the review focuses on the role of transcription and chromatin regulation in promoting DSB formation at this locus. 

The review is overall well written and timely, but I think some edits could make this interesting topic more generally accessible. In addition, some sections feel overly repetitive and could benefit from edits that remove redundancies.

 (Response) Thank you for your overall positive judgment. We revised all issues raised bellow.

  1. The link between transcription and DSB formation is indeed very strong at the ade6-M26, but I think the reader would benefit from highlighting that transcription does not always have a very clear causal relationship with DSB formation. For example, the HIS4 locus in budding yeast was initially shown to be very clearly linked to the activity of the transcription factors Bas1/2 but subsequent genome-wide analyses failed to observe a clear correlation between Bas1/2 activity and DSB formation.

 (Response) Thank you for this constructive comment. I added the descriptions about HIS4 locus and Bas1/2 in page 13.

  1. The discussion of ncRNA (section 6) should be tied better into the review. The relevance of these RNAs is not particularly well defined and the discussion of the data is quite technical. I suggest streamlining the discussion of the primary data and putting this data more into context with respect to the other aspects of ade6-M26 regulation discussed earlier. Also, how general is the role of ncRNAs in DSB formation? Some discussion of the roles of ncRNAs in DSB formation in other systems could be helpful to provide context.

 (Response) Thank you for this comment. I tied the discussion of ncRNA with several reviews to suggest the relevance of ncRNA in the regulation of event on chromosome such as transcription. The discussion of primary data was minimalized and the aspect related to regulation mechanisms of ade6-M26 discussed earlier was highlighted. The universal role of mlonRNA in the regulation of meiotic DSB formation was also discussed.

  1. Several pieces of information are presented multiple times, creating unnecessary redundancies. For example, the catalytic mechanism of Spo11/Rec12 is described three times (including in the abstract). Similarly, reintroduction of transcriptional effects at the beginning of section 6 seems unnecessary.

 (Response) Thank you. I agree to this comment and reduced the text for the mechanism of Spo11/Rec12 at section 5. I also reduced the text of transcriptional effects at the beginning of section 6.

  1. I suggest removing researchers’ names from the text to keep the focus on the science. The researchers’ contributions are readily seen in the citations.

 (Response) I agree to this comment and removed the researchers’ names from the text.

  1. In general, abbreviations should only be introduced for terms that are used again later. I may have missed it, but abbreviations like NGS and HAT do not seem to be used after their introduction. I also wonder whether less common abbreviations such as ADCRs are necessary. If there is no character limit then I think using “chromatin remodelers”, rather that ADCRs, throughout the later mentions would greatly help readability.

 (Response) Thank you. I agree to this comment. I omitted the used of these abbreviations.

  1. ade6-3049: is this allele a DSB hotspot? Some of the later discussion seems to imply this but this is not spelled out when the allele is introduced.

 (Response) I added the descriptions to indicate that ade6-3049 is hotspot.

  1. Clarify the term “naturally occurring M26 sequence sites”. At first read, I thought this meant that some natural S. pombe isolates encode the ade6-M26 allele. Later in the review it becomes clearer that this term refers to other instances of this sequence motif in the S. pombe genome.

 (Response) I agree this comment and added a description ‘(M26 heptanucleotide sequence occurring in wild type cells)’

  1. Figure 3B: Why are Atf1/Pcr1 omitted in the last panel? Is there any evidence the two proteins ever leave?

 (Response) Thank you. I fixed this error.

  1. Figure 4 and last lines in section 5: I am not convinced that there is any evidence for stable binding of Spo11/Rec12 to the DSB site. The observed lack of cleavage at the M26 sequence could simply be the result of a footprint by Atf1/Pcr1 that blocks access for the Spo11/Rec12 nuclease.

 (Response) Thank you. I fixed this error.

  1. The author makes a good attempt to simplify nomenclature by using the most commonly used gene names across organisms (e.g. Spo11 vs Rec12). However, for Spo11/Rec12 this nomenclature is not use consistently, and Red12 is introduced as the homolog of Spo11 twice. I think the species-specific nomenclature could be introduced earlier, using Rec12 from then on out. Alternatively, using a double name such as Spo11/Rec12 or Spo11 (Rec12) consistently could help avoid confusion.

 (Response) We agree to this comment. After the definition of Rec12 as a homolog of Spo11 in fission yeast, we used the term ‘Spo11/Rec12’ to indicate Spo11.

Minor points:

  1. When talking about Spo11/Rec12, please clarify that the similarity is specifically with topoisomerase II (or topoisomerase II-like enzymes), rather than topoisomerase in general.

 (Response) Thank you. I described that the similarity is specifically with topoisomerase II.

  1. Tyr should also be written out as tyrosine at first mention.

 (Response) Thanks. I fixed.

  1. Typo: Introduction, line 3: “Meiosis consists of DNA replication followed by two….”

 (Response) Thanks. I fixed.

  1. Introduction, paragraph 1, last line: crossovers and chiasmata should be plural in this sentence

 (Response) Thanks. I fixed.

  1. Introduction, paragraph 2, second and third sentences from the end: “resulting in gene conversion” and “results in gene conversion” is repetitive. I suggest deleting this clause in the first of the two sentences.

 (Response) Thanks. I fixed these points.

  1. Introduction, paragraph 2, last line: in S pombe there is clear evidence for single Holliday junctions. I suggest rephrasing this part and maybe referring to joint molecules (as a more general term) and then clarifying that these can be either double or single-Holliday junctions

 (Response) Thank you so much for this constructive comment. I added the descriptions to explain the single Holliday junction mediated CO.

Reviewer 2 Report

The review titled “Regulation mechanisms of meiotic recombination revealed from the analysis of a fission yeast recombination hotspot ade6-M26” provides an interesting and complete data revision regarding the  mechanisms involved in the meiotic recombination. It is well written and support a discussion of the already reported papers in the field. It will definitely be a good contribution to the field. Although the paper clearly is based in the knowledge provided by yeast data, the author could mention or discuss a little more about the existent data and knowledge from other animal models.

It would also improve the manuscript if the figures include a wider spectrum of colors. The current colors are in almost the same gamma and are enriched in yellow and grey. By increasing the spectrum of colors in the figures, the author will definitely help to deliver a more attractive and easier to follow concept to the readers

Not quite sure what is the aim of the chromatin paragraph in page 4. Although chromatin configuration is an essential process to repair DSBs and to define a CO region, the author only describes a quite broad information that seems not related to the aim of the paper and to the flow of the manuscript. I strongly suggest to improve it or erase it, even more by considering that a full paragraph about the chromatin mechanisms is written in page 6.

Author Response

Reviewer 2

The review titled “Regulation mechanisms of meiotic recombination revealed from the analysis of a fission yeast recombination hotspot ade6-M26” provides an interesting and complete data revision regarding the mechanisms involved in the meiotic recombination. It is well written and supports a discussion of the already reported papers in the field. It will definitely be a good contribution to the field. Although the paper clearly is based in the knowledge provided by yeast data, the author could mention or discuss a little more about the existent data and knowledge from other animal models.

 (Response) Thank you for your overall positive judgment. I agree to this comment and I added several descriptions about yeasts, arabidopsis and animal models.

It would also improve the manuscript if the figures include a wider spectrum of colors. The current colors are in almost the same gamma and are enriched in yellow and grey. By increasing the spectrum of colors in the figures, the author will definitely help to deliver a more attractive and easier to follow concept to the readers

 (Response) Thank you for this comment. I revised the all figures using a wider spectrum of colors.

Not quite sure what is the aim of the chromatin paragraph in page 4. Although chromatin configuration is an essential process to repair DSBs and to define a CO region, the author only describes a quite broad information that seems not related to the aim of the paper and to the flow of the manuscript. I strongly suggest to improve it or erase it, even more by considering that a full paragraph about the chromatin mechanisms is written in page 6.

 (Response) Thank you for this comment. I agree to your opinion that these sentences significantly overlapped with the descriptions in the section 4, and I erased these most descriptions from page 4.